# Novel Solution for Using Neural Networks for Kidney Boundary Extraction in 2D Ultrasound Data

**DOI:** 10.3390/biom13101548

**Published:** 2023-10-19

**Authors:** Tao Peng, Yidong Gu, Shanq-Jang Ruan, Qingrong Jackie Wu, Jing Cai

**Affiliations:** 1School of Future Science and Engineering, Soochow University, Suzhou 215006, China; 2Department of Health Technology and Informatics, The Hong Kong Polytechnic University, Hong Kong, China; 3Department of Radiation Oncology, UT Southwestern Medical Center, Dallas, TX 75390, USA; 4Department of Medical Ultrasound, Suzhou Municipal Hospital, Suzhou 215000, China; guyidong@njmu.edu.cn; 5Department of Electronic and Computer Engineering, National Taiwan University of Science and Technology, Taipei City 10607, Taiwan; sjruan@mail.ntust.edu.tw; 6Department of Radiation Oncology, Duke University Medical Center, Durham, NC 27710, USA; jackie.wu@duke.edu

**Keywords:** ultrasound kidney segmentation, deep fusion learning network, automatic searching polygon tracking, mathematical mapping model

## Abstract

*Background and Objective*: Kidney ultrasound (US) imaging is a significant imaging modality for evaluating kidney health and is essential for diagnosis, treatment, surgical intervention planning, and follow-up assessments. Kidney US image segmentation consists of extracting useful objects or regions from the total image, which helps determine tissue organization and improve diagnosis. Thus, obtaining accurate kidney segmentation data is an important first step for precisely diagnosing kidney diseases. However, manual delineation of the kidney in US images is complex and tedious in clinical practice. To overcome these challenges, we developed a novel automatic method for US kidney segmentation. *Methods*: Our method comprises two cascaded steps for US kidney segmentation. The first step utilizes a coarse segmentation procedure based on a deep fusion learning network to roughly segment each input US kidney image. The second step utilizes a refinement procedure to fine-tune the result of the first step by combining an automatic searching polygon tracking method with a machine learning network. In the machine learning network, a suitable and explainable mathematical formula for kidney contours is denoted by basic parameters. *Results*: Our method is assessed using 1380 trans-abdominal US kidney images obtained from 115 patients. Based on comprehensive comparisons of different noise levels, our method achieves accurate and robust results for kidney segmentation. We use ablation experiments to assess the significance of each component of the method. Compared with state-of-the-art methods, the evaluation metrics of our method are significantly higher. The Dice similarity coefficient (DSC) of our method is 94.6 ± 3.4%, which is higher than those of recent deep learning and hybrid algorithms (89.4 ± 7.1% and 93.7 ± 3.8%, respectively). *Conclusions*: We develop a coarse-to-refined architecture for the accurate segmentation of US kidney images. It is important to precisely extract kidney contour features because segmentation errors can cause under-dosing of the target or over-dosing of neighboring normal tissues during US-guided brachytherapy. Hence, our method can be used to increase the rigor of kidney US segmentation.

## 1. Introduction

Kidney segmentation algorithms are routinely used to extract regions of interest (ROIs) from entire medical images and help radiologists make clinical decisions [1]. Given its advantages of being painless, noninvasive, and cost-efficient, ultrasound (US) imaging is a good option for evaluating kidney health [2]. However, manually labeling US kidney images is tedious and complex. To reduce the workload of radiologists and increase the efficiency of annotation, there is a demand for an automatic US kidney segmentation algorithm for clinical applications [3]. It is challenging to develop such an algorithm because (1) the kidney boundary may not always be complete and prominent due to interference from neighboring tissues (e.g., intestinal gas) [4]; (2) the kidney boundary may have low contrast; (3) the intensity of the kidney structure may follow different distributions; and (4) kidney shape varies across patients. The challenges of kidney segmentation in trans-abdominal US images are illustrated in Figure 1.

Two main kinds of medical image segmentation algorithms are currently used: region-based [5,6] and contour-based [7,8]. Torres et al. [9] proposed a fast phase-based method for US kidney segmentation and achieved a DSC of around 0.81. Nevertheless, when a phase-based feature detection algorithm was utilized to extract the initial contour around the external dark-to-bright transition, the refinement step, which used a B-spline active surface framework, was prone to becoming stuck at a local minimum. Chen et al. [10] designed a deep convolutional neural architecture for segmenting US kidney slices. However, the outcome of the model was strongly influenced by the image quality, vagueness of the outline, and heterogeneous construction. This increased the potential of the method to wrongly detect and segment several slices. Yin et al. [11] combined a boundary detection network with a transfer learning architecture for US kidney segmentation. Their method may have been restrained by the capability of the transfer learning architecture to capture kidney features. In other words, the capability of their proposed method relied on the performance of the pre-trained visual geometry group (VGG) model. Unlike region segmentation algorithms, contour segmentation algorithms have the merit of easily detecting the appearance of an anatomical structure.

Using the shape representation [12] or the curve approximation method [13,14], contour-based segmentation algorithms can be used to delineate the contours of organs. Zheng et al. [15] integrated image intensity information and texture information into a dynamic graph-cutting method to segment US kidney slices. Their method achieved good segmentation performance with reasonable initialization (i.e., intensity and texture information). Marsousi et al. [16] proposed a new shape model to segment US kidney images using shape and anatomical knowledge as prior initialization. In Ref. [17], a phase-based distance regularized evolution model was proposed for segmenting the ROI of the kidney, with the partial phase and feature used as priors to improve segmentation accuracy. However, their method required too many parameters to be initialized by humans. Using a parametric super-ellipse as a global shape initialization, Huang et al. [12] designed a contour-based model for US kidney segmentation tasks with the aim of solving two problems: finding the segmented boundary of a fixed prior shape and determining the deformation parameters of the super-ellipse for the obtained segmented boundary. However, in practical scenarios, it was difficult for radiologists to manually determine an initialized shape-based ellipse’s central point that had the same location as the actual kidney contour’s central point.

We developed an automatic coarse-to-refinement segmentation network for kidney segmentation in US images. In the coarse segmentation stage, we used a deep fusion learning network (DFLN). In the DFLN, we integrated a deep parallel architecture consisting of an attention gate (AG) module [18] and a squeeze and excitation (SE) module [19] into the U-Net architecture [20]. Furthermore, we used an automatic searching polygon tracking (ASPT) method coupled with an adaptive learning-rate backpropagation neural network (ABNN) [21] to express a mathematical map function of a smooth kidney contour and optimize the coarse segmentation outcome. Compared with existing segmentation strategies, our new method has the following advantages:Our fully automatic coarse-to-refinement segmentation method is more suitable for practical applications than manual and semi-automatic methods that require excessive human intervention.Our previous study proposed a closed polygonal segment method [22] for the first time to address the drawbacks of the *K*-segments polygonal segment algorithm [23], which could not handle closed data well. Furthermore, we devised an enhanced polygonal segment algorithm [24,25] to minimize interference with the principal curve (PC) by abnormal data points. The above methods are standard PC-based methods. In this study, we devised an ASPT method to replace the standard PC-based methods for automatically determining vertices and clusters without prior knowledge.In our method, an explainable mathematical mapping function of the kidney boundary is represented by the parameters of the ABNN. During ABNN training, model deviation is reduced to yield a precise outcome.

Work related to the current study was accepted at the 2021 IEEE International Conference on Bioinformatics and Biomedicine (BIBM) conference [14]. There are some differences between the conference paper and the current study:We include more comprehensive literature in the current study.Compared with the conference work, which described a semi-automatic segmentation algorithm [14], we propose here a fully-automatic segmentation algorithm whose performance will not be affected by the selection of initial prior points.Unlike conventional PC-based algorithms [14,23], our ASPT algorithm has the advantage of being a mean shift clustering (MSC)-based method. This allows it to automatically determine the number of vertices and clusters without any human intervention. To the best of our knowledge, the current study is the first to propose a PC-based ASPT algorithm.Given the gradient vanishing issue arising from the backpropagation neural network used in our previous study [14], we used a rectified linear unit (ReLU) function [26] to replace the sigmoid function in the current study [27].In the conference work, we compared our method with hybrid methods such as deep belief network-closed polygon tracking (DBN-CPT) [22] and with deep learning methods such as mask region-based convolutional neural network algorithms [28] and Unet++ [29]. In the current study, we compared more recent methods such as A-LugSeg [7] and deep learning methods such as Transformer [30]. The A-LugSeg [7] method is a fully automated method previously developed by our group for lung segmentation.

## 2. Methods

### 2.1. Problem Formulation

As the vertices and clusters of traditional PC methods are manually predetermined [31], the aim of our method was to obtain an accurate kidney contour without manual intervention. First, given the ability of deep learning techniques to automatically learn from image features (i.e., ROI location, contour, and intensity), we used deep learning-based models for coarse segmentation. Then, the contour vertex set of coarse segmentation, *P_iv_* = {*p*_1_, *p*_2_,…, *p_iv_*}, was used as the input for polygon tracking-based methods. From this, the vertex sequence *D* = {*d*_1_, *d*_2_,…, *d_iv_*}∈*R^d^* = {(*t_i_*, (*v_x_*, *v_y_*)), *i* = 1, 2, …, *iv*, 0 ≤ *t*_1_ < *t_i_* < *t_iv_* ≤ 1} can be obtained, where *t_i_* is the sequence number of vertices and *p_i_*(*v_x_*, *v_y_*) represents the coordinates of the corresponding vertices, where *v_x_* and *v_y_* are the corresponding *x*- and *y*-axis coordinates, respectively, of vertex *v_i_*. However, the contour obtained by the polygon tracking-based method consisted of several segments. Next, vertex sequence *D* was used as the input for the neural network. The model error can be minimized during neural network training. Finally, we devised a mathematical formula (explained through the basic parameters of the neural network) to express a smooth kidney contour, as shown in Equation (1). For clarity, please see Table A1 in the Appendix A, which lists the abbreviations for all the notation used.
(1)$$f(t)=\left(v_{x}(t),v_{y}(t)\right)=\left(g(v_{x}(t)),g(v_{y}(t))\right)$$
(2)$$\left(g(v_{x}(t)),g(v_{y}(t))\right)=(e^{\sum_{i=1}^{s}\frac{1}{1+e^{-(tw1_{i}-m_{i})}}w_{2}{}_{i,1}-u_{1}},e^{\sum_{i=1}^{s}\frac{1}{1+e^{-(tw1_{i}-m_{i})}}w_{2}{}_{i,2}-u_{2}})$$

In Equations (1) and (2), *g*(•) is the value of the output units, *s* represents the number of hidden neurons, and *m* and *u* are hidden and output thresholds, respectively. *w*_1_ and *w*_2_ are the hidden and output weights, respectively.

There were several challenging issues to consider. First, deep learning networks receive large amounts of interest for automatic segmentation tasks. However, because of the poor contrast between kidneys and the surrounding tissue, selecting an optimal deep learning model for segmenting US kidney slices is not easy. Second, the main steps of current PC-based methods [7,32] are projection, vertex optimization, and vertex addition, in which the number of vertices is determined by human intervention. In our study, it was also challenging to identify the best method to automatically determine the vertices to describe the segmentation contour. Third, contours obtained using the PC method have several line segments. Identifying the best mathematical map function to express a smooth kidney contour is therefore difficult.

### 2.2. Overview of the Proposed Method

Our coarse-to-refinement-based method had two main components: (1) a coarse segmentation step and (2) a refinement step. The flowchart of our approach is shown in Figure 2. Stage 1 represents the deep fusion learning-based coarse segmentation stage, and Stage 2 is the refinement stage. In addition, the input and output of each stage of our developed system are displayed in Table 1, and all the notations are defined in the Appendix A.

### 2.3. Coarse Segmentation Step

Our work used a DFLN for coarse kidney segmentation, where the DFLN model comprises an SE module [19], an AG module [33], and a U-Net architecture [20]. As the SE and AG modules have a good ability to highlight salient features, we used both to generate different fusion variants. Figure 3 shows the structure of four fusion variants (two serial and two parallel). The combination of the AG and SE modules occurs at each skip connection of the U-Net structure.

With the U-Net structure as the backbone, we used one serial architecture of AG followed by SE (Serial 1) and another serial architecture of SE followed by AG (Serial 2). Furthermore, using different inputs for the SE module, we used different parallel architectures. The biggest difference between the two parallel architectures lies in the input for the SE: one uses features from decoding paths (Parallel 1) and the other uses features from encoding paths (Parallel 2).

The U-Net structure contains encoder and decoder blocks. The SE module contains a squeeze module (global max-pooling layer) and an excitation module (two 1 × 1 convolutions followed by a ReLU [26] and sigmoid function [34]). In addition, the AG module contains a concatenation between the output of the corresponding down-sampled and up-sampled layers, each followed by the ReLU, a 1 × 1 convolution, a sigmoid function, and a multiplication operation.

### 2.4. Acquire Vertex Sequence

In this section, the ASPT method is applied to obtain the vertex sequence. In contrast to standard PC-based methods [31], our proposed method leveraged the ability of the MSC-based method to automatically determine the numbers of vertices and clusters without any prior knowledge [35], resulting in the automatic generation of the PC. Figure 4 shows the difference between related PC-based and our ASPT methods.

#### 2.4.1. Normalization

To unify the dataset, the contour vertex set of the coarse segmentation result, *P_iv_*, is normalized to the range of {(−1, −1)~(1, 1)} based on the min-max normalization method [36].

#### 2.4.2. Adaptive MSC Method

The basic goal of the MSC-based method is to find cluster points based on different cluster sets that meet different probability density distributions. When the initial points converge at the cluster point with the maximum local probability density, these initial points are designated as belonging to the same cluster [37].

The traditional MSC method was proposed by Cheng et al. [37] for searching data clusters. However, many subsequent studies have demonstrated that MSC with adaptive bandwidth generates more accurate outcomes than an algorithm with fixed kernel bandwidth, particularly in high-dimensional feature space [38]. We used the adaptive MSC method [38], which combines the automatic selection of a kernel bandwidth scheme with MSC to automatically determine the numbers of vertices and clusters.

#### 2.4.3. Vertex Optimization Step

Standard PC-based vertex optimization step

The purpose of this step was to minimize the penalty distance function *G_iv_*(*v_i_*) so that the locations of all the vertices are optimized [31]. Let $\pi (v_{i})=r^{2}(1+\cos\gamma_{i})$, $\mu_{+}(v_{i})=\left|\right|v_{i}-v_{i+1}|{|}^{2}$, and $\mu_{-}(v_{i})=\left|\right|v_{i}-v_{i-1}|{|}^{2}$ in which *r* is defined as the data radius, represent the largest distance between point *p* and centroid *p* of *P_i_*′*_v_*, as shown in Equation (3).
(3)$$r=\max_{p\in P_{iv}}\left\|p-\frac{1}{iv}\sum {}_{p\prime \in P_{iv}}p\prime \right\|$$

Combining the *π*(*v_i_*), *μ*_+_(*v_i_*), and *μ*_−_(*v_i_*) functions, Kegl et al. [31] computed the curvature penalty *CP*(*v_i_*) on vertex *v_i_*, defined by Equation (4), where *a* denotes the number of vertices.

Let $\tau (v_{i})=\sum_{x\in V}\Delta (x,v_{i})$, $\sigma_{+}(v_{i})=\sum_{x\in S}\Delta (x,s_{i})$, and $\sigma_{-}(v_{i})=\sum_{x\in S}\Delta (x,s_{i-1})$, where *V* and *S* are the vertex and segment sets, respectively, and the average squared distance function Δ*_iv_*(*v_i_*) of *v_i_* is defined in Equation (4).
(4)$$\Delta_{iv}(v_{i})=\left\{\begin{array}{ll} \tau (v_{i})+\sigma_{+}(v_{i}) & \mathrm{if}\;i=1\\ \sigma_{-}(v_{i})+\tau (v_{i})+\sigma_{+}(v_{i}) & \mathrm{if}\;1<i<iv+1\\ \sigma_{-}(v_{i})+\tau (v_{i}) & \mathrm{if}\;i=a+1 \end{array}\right.$$

Penalized distance function *G_iv_*(*v_i_*) is defined in Equation (5), where the penalty metric *λ* is a positive number and $\lambda =\lambda^{\prime }\cdot \frac{b}{iv^{1/3}}\cdot \frac{\sqrt{\Delta_{iv}(f_{is,iv})}}{r}$; *λ*′ is the adjustment parameter with a constant value of 0.13 [23], and *iv* and *is* are the numbers of vertices and segments, respectively.
(5)$$G_{iv}(v_{i})=\frac{1}{n}\times \Delta_{iv}(v_{i})+\lambda \times \frac{1}{is+1}CP(v_{i})$$

2.Our modified vertex optimization step

In standard PC-based methods [31], the curvature penalty *CP*(*v_i_*) is calculated using a triangular-based function, as shown in Equation (3). To increase the efficiency of standard PC-based methods, we re-designed the penalized distance function of the vertex optimization step using the curvature penalty function *MP*(*v_i_*), with addition and averaging, to replace *CP*(*v_i_*). The newly designed penalized distance function *G*′*_iv_*(*v_i_*) is expressed as Equation (6), where *MP*(*v_i_*) is defined as a penalty imposed on the total curvature of the principal curve defined by Equation (7).
(6)$$G_{iv}^{\prime }(v_{i})=\frac{1}{iv}\times \Delta_{iv}(v_{i})+\lambda \times \frac{1}{k+1}\times MP(v_{i})$$
(7)$$MP(v_{i})=\frac{1}{is}\times (\sum_{i=1}^{is}\Delta (p,v_{i}))$$

### 2.5. Explainable Mathematical Map Function of the Kidney Contour

After executing the ASPT method, we obtained the vertex sequence *D*, consisting of the number of vertex sequences *t* and the relevant vertices’ coordinates *v_i_*(*x_i_*, *y_i_*). The kidney contour, comprising several segments, was simultaneously obtained. To smooth the contour, we used a three-layered ABNN for training. *t* was used as the input for the ABNN, and *v_i_*(*x_i_*, *y_i_*) was applied to reduce the mean square error [39]. The sigmoid function *h*_1_ = 1/(1 + *e*^−*x*^) was applied in the forward-propagation procedure of the ABNN from the input to the hidden layers. Concurrently, the ReLU function *h*_2_ = max{0, *x*} was used from the hidden to the output layers. After training, we identified a mathematical map formula (expressed by the basic parameters of the ABNN) so that the smooth kidney boundary indicated by the output of the neural network (i.e., the updated vertices) matched the ground truth (GT), as shown in Equations (1) and (2).

### 2.6. Materials

We evaluated our segmentation network on a kidney dataset obtained from the Suzhou Municipal Hospital (SMH), Suzhou, Jiangsu, China. The kidney dataset, named the SMH dataset, consists of trans-abdominal US scanning-based images from 115 patients without kidney disease. The SMH data were obtained using a Mindray DC-8 US system with a 1.3–5.7 MHz low-resolution linear transducer. The probe detection depth and frequency were set to 200 mm and 4 MHz, respectively, and the amplifier gain was within the range [3, 33 dB]. We used both axial and sagittal view kidney images from each patient for evaluation. Three professional physicians manually delineated all the images. The consensus GT was then decided based on the majority of the three physicians’ labels. Two common metrics, DSC [40] and the Jaccard similarity coefficient (Ω) [25], were used for evaluating the performance of our model.

We split the SMH data of 115 patients (1380 slices) into three groups, namely 80 patients (960 slices) for training, 12 patients (144 slices) for validation, and the remaining 23 patients (276 slices) for testing. We resampled all of the slices from the original resolution, 1200 × 900 pixels, to 600 × 450 pixels. To avoid overfitting, we increased the size of the training dataset using rotation within [−20°, 20°] until it reached a total of 2000 slices. When the DFLN was being trained, we used the Dice loss function to calculate loss and the Adam optimizer [17] to adjust the learning rate. In addition, the stochastic gradient descent method was applied to optimize the ABNN. The initial learning rate, momentum value, and maximum training epochs were 0.4, 0.9, and 1000, respectively.

## 3. Results

### 3.1. Comparison with Different Variants

Table 2 represents the testing outcomes of four variant models based on the DSC value. Figure 5 represents the visual qualitative results of the four variant models based on two randomly selected cases. Parallel architectures (Parallel 1 and 2) showed better performance than serial architectures (Serial 1 and 2), with DSC and Ω values of 0.89 and 0.899 and 0.805 and 0.813, respectively, for the two parallel architectures. However, compared with Parallel 1, the DSC and Ω values of Parallel 2 increased by 1.01% and 0.99%, respectively. Hence, we used Parallel 2 as the coarse segmentation step of our proposed method.

### 3.2. Ablation Experiments

Ablation experiments were conducted to assess the influence of different components of our method. The ablation results are shown in Table 3. Figure 6 reports the visual comparisons using two representative cases. Compared with Model 1, the DSC and Ω of other models (with the refinement step) increased by 1.55–5.22% and 8.11–14.8%, respectively. In addition, for Models 2–6, the DSC values were higher than 91% and the standard deviations were lower than 6.5%. This demonstrates that the proposed refinement step can fine-tune the coarse segmentation outcomes. All in all, our model shows optimal capability.

### 3.3. Robustness Evaluation of Our Proposed Method Using a Testing Set Corrupted by Gaussian Noise

We used testing images corrupted by various degrees of Gaussian noise to evaluate the performance of our proposed method. Different degrees of Gaussian noise can have dissimilar influences on the capability of a method [25]. In our experiment, the standard deviation (σ) was assigned a set value (i.e., 0, 10, 25, or 50). As shown in Table 4, compared with the case in which σ = 0, when σ rose from 10 to 25, to 50, the DSC value increased from 0.96% to 1.93% and to 3.61%, respectively, and the Ω decreased from 1.85% to 2.52% and to 3.77%, respectively. However, the mean DSC values for all cases with different levels of noise were greater than 91%. The qualitative results of a randomly picked case are displayed in Figure 7.

### 3.4. Comparison with Existing Fully Automated Techniques

We compared our methods with two existing techniques: hybrid [7] and deep-learning-based [28,29]. We used the SMH kidney US data for internal validation and additional trans-abdominal US kidney data for external validation. The newly added kidney data were provided by the Beijing Tsinghua Changgung Hospital (BTCH), Beijing, China. The kidney dataset, named the BTCH dataset, was obtained from 45 brachytherapy patients (total of 450 slices) using a HI VISION Avius L US and a 1–5 MHz convex array probe. The mechanical index was set to 0.8, with a probe detection depth of 15 mm and amplifier gain between 2 and 30 dB. For better assessment, we used the commonly used DSC and Ω as the evaluation metrics.

#### 3.4.1. Internal Evaluation of SMH Data

As shown in Table 5, all the hybrid techniques performed better than the deep learning techniques; the values of DSC and Ω were as high as 8.3% and 16.6%, respectively, showing that the refinement module can optimize the coarse segmentation outcomes. All in all, our method shows better performance (i.e., precision and robustness) than existing methods.

#### 3.4.2. External Evaluation of BTCH Data

The BTCH dataset contains 450 slices from 45 brachytherapy patients, with all the slices resized to 600 × 450 pixels. These slices were used to evaluate the performance of all the models. Table 6 shows the performance of all the state-of-the-art methods on the BTCH data for external evaluation. Compared with the deep learning models, the mean values of DSC and Ω of the hybrid models increased to as high as 7.87% and 16.1%, respectively. As external evaluation is more challenging, the performances of all the methods were slightly lower in the external evaluation (Table 6) than in the internal evaluation (Table 5). The DSC and Ω of our method attained maximum values of 93.2 ± 3.7% and 92.1 ± 4.6%, respectively.

## 4. Discussion

This paper presents a novel mechanism to segment kidneys in US images. The proposed method has three attributes: (1) an automatic deep fusion learning network; (2) an automatic searching polygon tracking method; and (3) an interpretable mathematical formula for the kidney boundary. To assess the capability of our method for kidneys with inconsistent appearances, kidney images of 115 patients were used for a detailed qualitative and quantitative evaluation. Our work demonstrated that (1) our model exhibited good outcomes for different patients and evaluation metrics (DSC and Ω); and (2) our model outperformed other existing models. We discuss more details of the overall work below.

We collected trans-abdominal US scanning-based data from 115 patients, with left and right kidney images for each patient. The left and right kidneys could not be acquired in the same image because (1) there is a long distance between them, (2) they are separated by the spine, and (3) the probe scan range is limited.

As shown in Table 2, different serial and parallel architectures resulted in different levels of performance for the coarse segmentation strategy. There are three aspects of the outcomes presented in Table 2 to be discussed. ***First***, overall, the parallel architecture showed better capability than the serial architecture. Both the AG [18] and SE [19] modules are known to have a good ability to boost relevant features and remove irrelevant features. However, using serial AG and SE modules may cause the deletion of a large amount of information, some of which may be useful. ***Second***, of the serial architectures evaluated, SE-AG (Figure 3b) showed better performance than AG-SE (Figure 3a). The main reason for this is that the AG module has a more complex structure than the SE module (as shown in Section 2.3), which makes it difficult to train the AG module and avoid the loss of meaningful information. ***Third***, as illustrated in Table 2, the Parallel 2 model performed better than the Parallel 1 model. The main difference between these two parallel architectures is in the input for the SE module, with one using features from the decoding path (Parallel 1) and the other using features from the encoding path (Parallel 2). Parallel 2 may perform better because using encoding features as the input for the SE module carries the merit of the SE module to emphasize meaningful features and suppress less useful features.

As shown in Figure 6, we used an ablation comparison to demonstrate the capability of our method. The regions indicated by arrows are missing or ambiguous due to different factors. The white and green arrows indicate the blurry boundary of the kidney caused by intestinal gas and the spleen, respectively. The orange arrow indicates the ambiguous boundary of the kidney due to the kidney’s thickness and internal structure (i.e., renal pelvis, calyces, blood vessels, and adipose tissue). However, the model with a refinement step still obtained highly accurate results.

As US images are grayscale, most of the pixels are black and have a gray value of zero. To better distinguish the effect of other pixel points (gray value > 0), we only selected the number of pixels within the range [0, 10,000] in the image to show the distribution of pixel points with different gray values (Table 4 and Figure 7). When we added more Gaussian noise (σ increases), the number of pixels with gray values greater than 0 increased. This illustrates that the degree of damage to the original images increased. However, the DSC values were greater than 90% even for images with a severe level of noise (σ = 50). This indicates that the blurry boundaries were efficiently detected (Table 3 and Figure 7). Figure 8 shows zoomed-in images corresponding to those in Figure 7.

Although our method yielded promising results, several aspects require improvement. First, we want to further evaluate our method on multiple modalities (i.e., computed tomography and magnetic resonance slices) and multi-site data. Second, to achieve the goal of real-time clinical applications, model compression of our coarse-to-refinement method may be necessary to reduce the memory burden during the execution process. Third, we wish to evaluate our method on different organs or multi-organs such as the prostate, bladder, and fetal head. Finally, chronic nephritis, renal vascular disease, kidney transplantation, hydronephrosis, kidney tumors, and other diseases require highly accurate measurements of kidney volume. This plays a crucial role in the selection of treatment methods and postoperative evaluation. In the future, we will discuss the performance of our method in these directions.

## Figures and Tables

**Figure 1 biomolecules-13-01548-f001:**
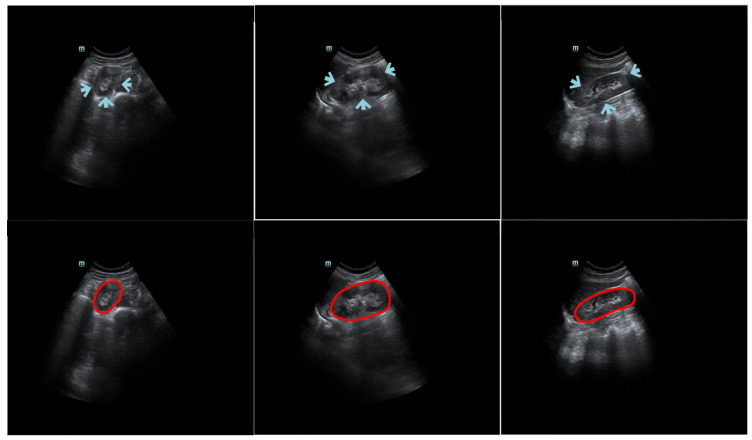
Example trans-abdominal US kidney images. Portions of the kidney boundary, pointed to by arrows, do not have high contrast with the surrounding tissue. The first column shows an axial image, and the second and third columns show sagittal images. The second row displays the corresponding ground truth of the original data in the first row. The red circles represent the contour manually delineated by the professional radiologists.

**Figure 2 biomolecules-13-01548-f002:**
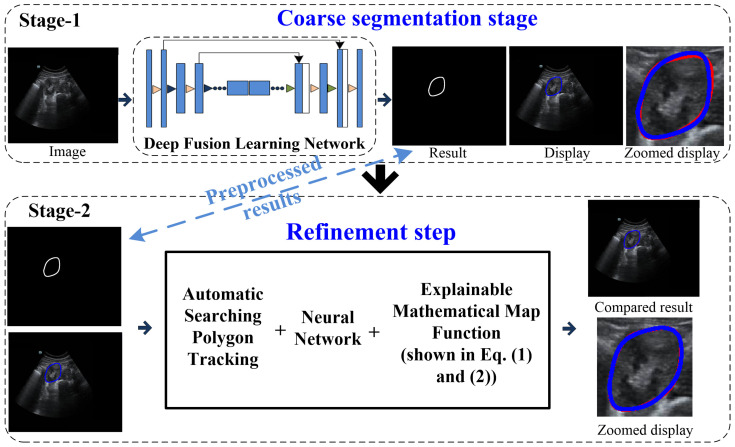
Flowchart of the coarse-to-refinement-based system. Blue contours represent the experimental results, and red contours represent the ground truth.

**Figure 3 biomolecules-13-01548-f003:**
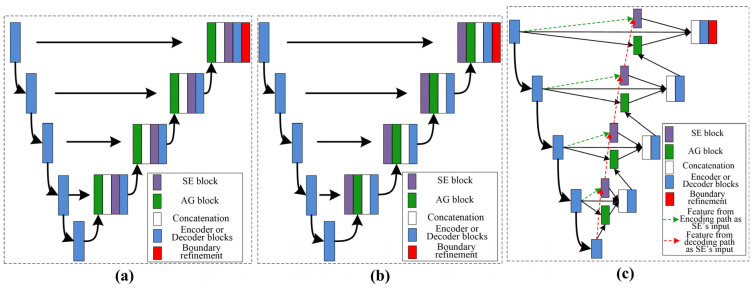
Description of the variants’ architecture, where (**a**) represent serial AG followed by SE, (**b**) represent the serial SE followed by AG, and (**c**) show both parallel architectures.

**Figure 4 biomolecules-13-01548-f004:**
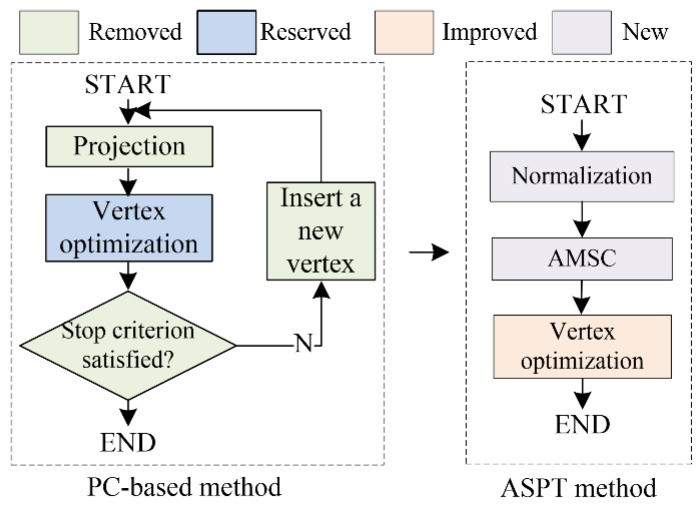
Difference between the standard PC-based method and our ASPT method. The main steps of the PC-based methods are projection, vertex optimization, and vertex addition. Unlike PC-based methods, our ASPT method only uses the newly added normalization and the adaptive MSC method while including a modified vertex optimization step.

**Figure 5 biomolecules-13-01548-f005:**
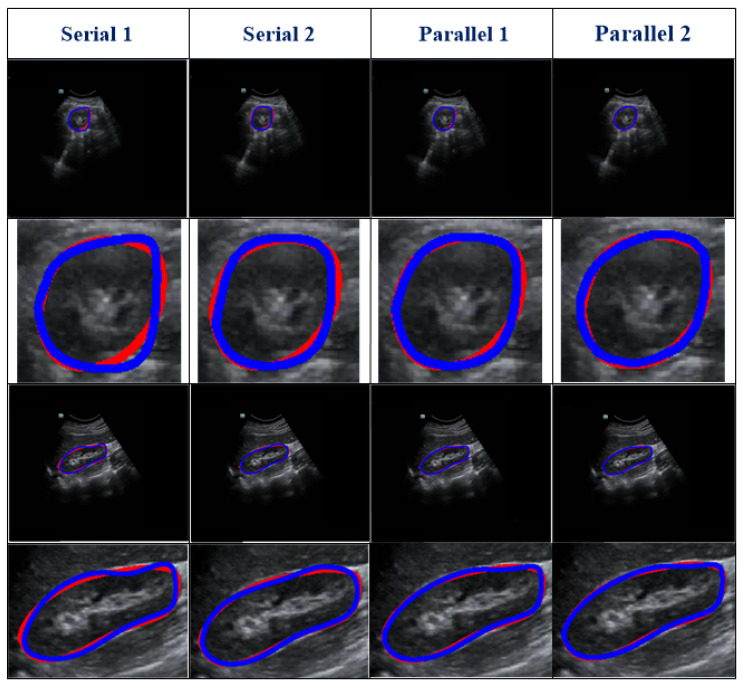
Visual results of four variant models. Two randomly selected cases are presented under different views. The blue and red lines show the experimental outcome and GT, respectively. The first two rows show the axial view, and the last two rows show the sagittal view. The first and third rows show the overlap between the segmentation result and the GT. The second and fourth rows show the corresponding zoomed-in display.

**Figure 6 biomolecules-13-01548-f006:**
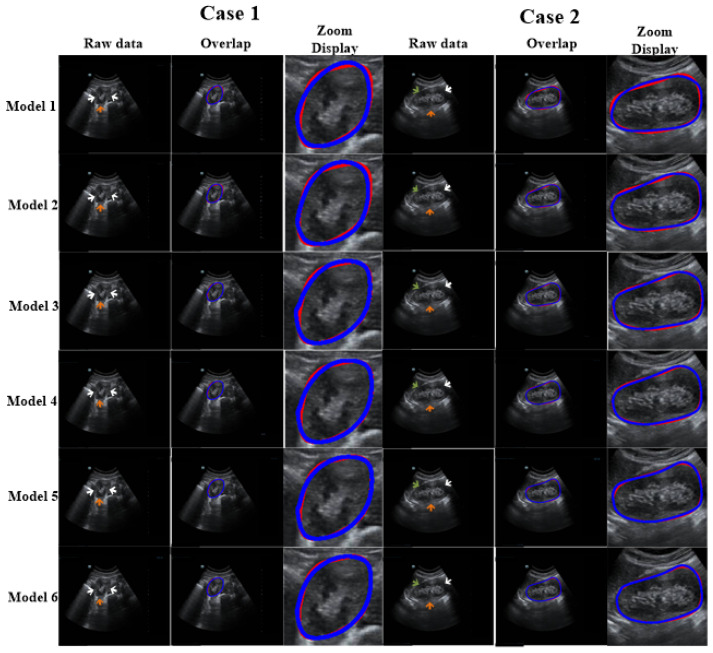
Discussion of the impact of each module of our proposed method. The arrows show missing or blurry boundaries. The blue and red curves represent the experimental outcome and GT, respectively.

**Figure 7 biomolecules-13-01548-f007:**
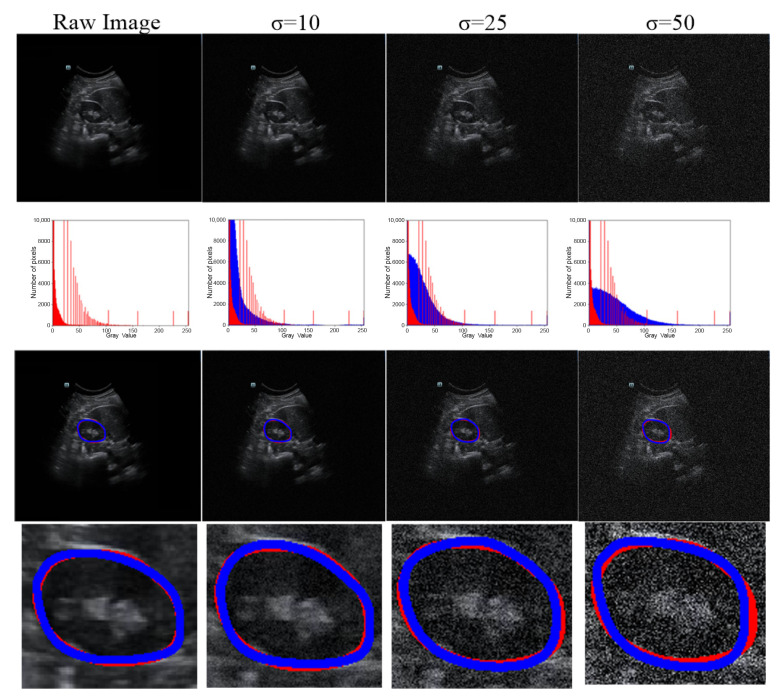
Robustness of our method under various degrees of Gaussian noise. Row 1, raw or noisy data; row 2, histograms of the raw or noisy images; row 3, the overlap between the GT and segmentation results; and row 4, corresponding zoomed-in results. The red line represents the GT, and the blue line represents the segmentation result.

**Figure 8 biomolecules-13-01548-f008:**
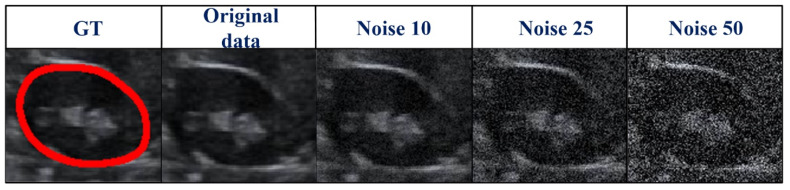
Zoomed-in images (Figure 7) corrupted by various degrees of Gaussian noise. GT, ground truth.

**Table 1 biomolecules-13-01548-t001:** Input/output of each stage of our developed system.

Model	Input	Output
DFLN	Original data	Coarse segmentation outcome (contour vertices set *P_iv_*)
ASPT	Coarse segmentation outcome (contour vertices set *P_iv_*)	Vertices sequence *D* (sequence number of vertices *t* and their corresponding coordinates *p*)
ABNN	Vertices sequence *D*	Refined result

DFLN: deep fusion learning network; ASPT: automatic searching polygon tracking; ABNN: adaptive learning-rate backpropagation neural network.

**Table 2 biomolecules-13-01548-t002:** Differences among different models. Serials 1 and 2 use serial architecture of AG followed by SE and SE followed by AG, respectively. Parallel 1 and 2 denote parallel architecture, using features from the decoding and encoding paths, respectively, as the input for the SE. The details of the four models are shown in Figure 3.

	DSC ± SD (%)	Ω ± SD (%)
Serial 1	87.1 ± 9.2	78.6 ± 12.1
Serial 2	87.9 ± 8.9	79.7 ± 11.1
Parallel 1	89.0 ± 8.1	80.5 ± 8.8
Parallel 2	89.9 ± 6.9	81.3 ± 8.7

**Table 3 biomolecules-13-01548-t003:** Ablation results. Except for Model 1, all the methods are coarse-to-refinement structures. DSC, dice similarity coefficient; SD, standard deviation; Ω, Jaccard similarity coefficient; P2: Parallel 2; MSC: mean-shift clustering; AMSC: adaptive mean shift clustering; VOS: vertex optimization step; MVOS: modified vertex optimization step; BNN: backpropagation neural network; ABNN: adaptive learning-rate backpropagation neural network.

	Architecture	DSC ± SD (%)	Ω ± SD (%)
Model 1	P2	89.9 ± 6.9	81.3 ± 8.7
Model 2	P2+MSC+VOS+BNN	91.3 ± 4.9	87.9 ± 6.4
Model 3	P2+AMSC+VOS+BNN	92.1 ± 4.5	89.2 ± 5.6
Model 4	P2+MSC+MVOS+BNN	92.4 ± 4.6	89.9 ± 6.1
Model 5	P2+AMSC+MVOS+BNN	93.8 ± 3.8	90.8 ± 5.3
Model 6	P2+AMSC+MVOS+ABNN	94.6 ± 3.4	93.4 ± 4.1

**Table 4 biomolecules-13-01548-t004:** Quantitative outcomes of our method under various degrees of Gaussian noise. σ = 0 denotes no noise added to the raw image.

	DSC ± SD (%)	Ω ± SD (%)
Raw set (σ = 0)	94.6 ± 3.4	93.4 ± 4.1
σ = 10	93.7 ± 3.9	91.7 ± 4.6
σ = 25	92.8 ± 4.2	91.1 ± 4.9
σ = 50	91.3 ± 4.7	90 ± 5.4

**Table 5 biomolecules-13-01548-t005:** Comparison with existing fully automatic techniques on SMH data (DSC and Ω in mean ± standard deviation [SD]).

Reference	Method	Technique/Model	DSC ± SD (%)	Ω ± SD (%)
[28]	Mask-RCNN	Deep learning (fully automatic)	87.3 ± 7.2	80.1 ± 9.2
[30]	Transformer	Deep learning (fully automatic)	89.4 ± 7.1	80.9 ± 8.7
[7]	A-LugSeg	Hybrid (fully automatic)	93.7 ± 3.8	91.6 ± 4.9
Current	Our method	Hybrid (fully automatic)	94.6 ± 3.4	93.4 ± 4.1

**Table 6 biomolecules-13-01548-t006:** Comparison with existing fully automated techniques on BTCH data (DSC and Ω in mean ± standard deviation [SD]).

Reference	Method	Technique/Model	DSC ± SD (%)	Ω ± SD (%)
[28]	Mask-RCNN	Deep learning (fully automatic)	86.4 ± 7.5	79.3 ± 9.5
[30]	Transformer	Deep learning (fully automatic)	88.1 ± 7.3	79.8 ± 8.9
[7]	A-LugSeg	Hybrid (fully automatic)	92.5 ± 4.2	90.6 ± 5.4
Current	Our method	Hybrid (fully automatic)	93.2 ± 3.7	92.1 ± 4.6

## Data Availability

Data will be made available on reasonable request.

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
