# Peer review of "Novel Solution for Using Neural Networks for Kidney Boundary Extraction in 2D Ultrasound Data"

_biomolecules, 2023, doi:10.3390/biom13101548_

Round 1
Reviewer 1 Report
It is not a well-written manuscript. I provided some examples here:
Line 107-110:
“Related work in our current study was accepted at the 2021 IEEE International Con- 107 ference on Bioinformatics and Biomedicine (BIBM) conference, whose acceptance rate was 108 19.6% [11], where our previous work was picked for oral presentation. There are some 109 differences between our previous conference paper and the current study, shown below, “
Line 111
“We include more comprehensive literature in our current study. ”
Line 122-125
“A more comprehensive comparison with our previous conference work [11] is pre- 122 sented. In our previous conference work, we compared our method with hybrid 123 methods, including CPL-BNN [19] and DBN-CPL [21], and deep learning methods, 124 including Mask-RCNN [27] and Unet++ [28]. ”
….
I advise rejecting the manuscript.
It is not a well-written manuscript. I provided some examples here:
Line 107-110:
“Related work in our current study was accepted at the 2021 IEEE International Con- 107 ference on Bioinformatics and Biomedicine (BIBM) conference, whose acceptance rate was 108 19.6% [11], where our previous work was picked for oral presentation. There are some 109 differences between our previous conference paper and the current study, shown below, “
Line 111
“We include more comprehensive literature in our current study. ”
Line 122-125
“A more comprehensive comparison with our previous conference work [11] is pre- 122 sented. In our previous conference work, we compared our method with hybrid 123 methods, including CPL-BNN [19] and DBN-CPL [21], and deep learning methods, 124 including Mask-RCNN [27] and Unet++ [28]. ”
….
I advise rejecting the manuscript.
Author Response
Dear Reviewer,
Thanks for your valuable comments. The spelling errors, syntax errors, and seemingly unfinished sentences have been checked and corrected by the professional English editing service. Please see the attachment, which is the point-to-point response.
Best,
Tao Peng

Reviewer 2 Report
In this study, the authors present a new method to segment kidneys in ultrasound images. It remains challenging to automatically segment kidneys in clinical ultrasound images due to the kidneys’ varied shapes and image intensity distributions, although semi-automatic methods have achieved promising performance. This paper ambitiously introduce a novel approach based on deep fusion learning algorithms, an automatic searching polygon tracking method and a novel approach to find the kidney boundary. The study data included of 115 data from patients, and data were sub-divided in training data, validation data and testing data. Evaluation metrics included the Dice Similarity Coefficient (DSC), which is a similarity metric commonly used in image segmentation. The authors showed convincingly that their method had an increased DSC value compared to other methods, around 90-95% which is very impressive. They also demonstrated in their testing phase that adding noise did not significantly affect the outcome (DSC values). In general, the study is very well conducted and the authors should be congratulated for a very detailed description of the method, and this novel approach may have significant impact in the way ultrasound images of kidneys are analyzed. Besides, I should congratulate the authors for their detailed Introduction section, and in particular, I acknowledge the authors for their descriptions how the methods have been approved since their last public dissemination (conference).
Some minor comments:
The Results section starts with some basic methods information – perhaps this should be moved to the Methods section?
There is a lack of information about the patients? What kind of patients? Kidney patients? If patients did not have kidney diseases, then it would be a good idea to discuss future applications in kidney disease patients. If data originated from kidney patients, then some comments should be written whether the proposed method provided equally DSC scores between healthy and disease kidneys. Also, it is not clear to me (in the Introduction section), in which kidney diseases there is a need for very precise measurements of kidney volumes, which could become essential for therapeutic decisions - perhaps the authors could elaborate?
The provided data were all recorded from the same ultrasound system. Is this system a standard clinical system comparable to most other clinically ultrasound systems?
Author Response
Dear Reviewer,
Thanks for your valuable suggestions. Please see the attachment, which is the point-to-point response.
Best,
Tao Peng

Reviewer 3 Report
The manuscript presents a novel approach to automating kidney segmentation in ultrasound imaging, with the following key points: Kidney segmentation in ultrasound is crucial for early detection of kidney stones and renal masses. Manual delineation is currently the standard practice, but is laborious and time-consuming. The proposed method consists of three sequential steps for ultrasound kidney segmentation. First: employs a deep fusion learning network to initial coarse segmentation. The second step refines the initial result using an automatic polygon tracking method and a machine learning network that defines the kidney contour through mathematical formulas. The method was evaluated on a dataset comprising 1,380 trans-abdominal ultrasound images from 115 patients and compared to state-of-the-art methods.
One concerns about the dataset: Is there any ethical number issued for using the patient dataset?
It is advisable to consider excluding some citations from the author's paper, such as references 19 and 30, which appear to be less relevant to the current study.
Author Response
Dear Reviewer,
Thanks for your constructive suggestions. Please see the attachment, which is the point-to-point response.
Best,
Tao Peng

Reviewer 4 Report
This study sought to perform automatic kidney segmentation on ultrasound using a multi-step process. A few variant architectures were assessed on a population of 115 patients with 1,380 ultrasound images of the kidneys. The model performed well compared to the ground truth on this dataset. The authors also perform ablation experiments, as well assessing the effect of Gaussian noise. A few points:
Abstract: Perhaps instead of boring, consider using other descriptors, such as time-consuming and tedious.
Abstract: “Protection of risk strictures”: I am not sure what you mean by this.
Abstract: “According to comprehensive comparisons on different noise levels, our method achieved accurate and robust results in prostate segmentation.” The abstract is talking about kidney segmentation rather than prostate segmentation.
Abstract: “Hence, our method acquires the capacity to increase the outcome of the diagnosis of kidney disease.” Segmenting kidneys is not the same thing as diagnosing kidney disease, for which performance metrics are not given. Perhaps you could state something about how this your algorithm could be an important first step for diagnosing various kidney diseases by providing an accurate segmentation of the kidneys.
Introduction, Line 47: I am not sure that I would say that the kidney boundary is always missing.
Introduction, Line 50: “Several challenging kidney segmentation cases in transabdominal ultrasound images are shown in Fig. 1.” Are these different cases or different orientations of the same patient?
Figure 1: I am not sure if I would characterize these as examples were the boundary is truly missing, as I would imagine a radiologist would not have a problem in drawing the boundary. Perhaps one can say that portions of the boundary of the kidney do not have high contrast with the surrounding tissues, which could present a challenge for segmentation algorithms.
Introduction, Line 57: “the refinement step using the B-spline active surface framework can always be stuck at a local minimum.” Is the technique prone to becoming stuck at a local minimum or does it truly always get stuck there?
Introduction, Line 82: “However, the pre-set information that uses the same central points between initialized shape-based ellipse information and the kidney is not good.” Consider briefly explaining why it is not good.
Introduction, Line 107: “Related work in our current study was accepted at the 2021 IEEE International Conference on Bioinformatics and Biomedicine (BIBM) conference, whose acceptance rate was 19.6% [11], where our previous work was picked for oral presentation.” I am not sure that mentioning the acceptance rate of the conference and mentioning that the work was accepted for oral presentation is relevant here.
Table 1: Consider spelling out the full term for the DFLN, ASPT, and ABNN acronyms under the table, so the reader can quickly peruse the tables without directly searching for them in the text.
Introduction, Line 173: “Fig. 3 shows the structure of four different fusion variants (two serial and parallel variants).” To clarify, does Figure 3c show both parallel architectures?
Figure 4: Consider modifying the graphic so that “Termination criterion satisfied” is not outside the diamond graphic.
Results: What was the medical specialty of the three physicians? Are they medical trainees or attending physicians?
Results: Did this study have approval by an institutional review board (IRB)?
Implementation Details: Were slices for the same patient in different sets (i.e. some slices for patient 1 were in training, some were in validation, and some were in test)? Please clarify if this is not the case. If so, that could overstate model performance.
Robustness evaluation, Line 299: “As shown in Table 4, compared with the case in which σ = 0, when σ raised from 10 to 25, to 50, the DSC value increased from 0.96% to 1.93%, to 3.61%, respectively”: It seems like the addition of Gaussian noise decreased the DSC.
Table 5: It appears that studies from the literature were compared to your method on the dataset you used to develop the model. It would have been ideal if there was also an external validation dataset.
Appendix: Consider referencing Table 6 when the equations are first introduced for the sake of clarity.
This study could benefit from some editing of the grammar.
Author Response
Dear Reviewer,
Thanks for your valuable suggestions that can improve the quality of this manuscript. Please see the attachment, which is the point-to-point response.
Best,
Tao Peng

Round 2
Reviewer 4 Report
The authors have substantially improved the paper. It is now much easier to read, and my concerns were addressed. In particular, the inclusion of the external validation set is helpful in establishing the generalizability of the technique. My one comment is that just as the SMH data is mentioned in the Methods, the new external validation BTCH dataset should also be briefly mentioned in the Methods (it is introduced in the Results).